# Lossless Watermarking Algorithm for Geographic Point Cloud Data Based on Vertical Stability

**Mingyang Zhang [1], Jian Dong [1,2,*], Na Ren [3,4] and Shuitao Guo [3]**

[1]  Department of Military Oceanography and Hydrography & Cartography, Dalian Naval Academy,
     Dalian 116018, China; navyzhmy@163.com
[2]  Key Laboratory of Hydrographic Surveying and Mapping of PLA, Dalian Naval Academy,
     Dalian 116018, China
[3]  Key Laboratory of Virtual Geographic Environment, Nanjing Normal University, Ministry of Education,
     Nanjing 210023, China; 09359@njnu.edu.cn (N.R.); 201302105@njnu.edu.cn (S.G.)
[4]  Hunan Engineering Research Center of Geographic Information Security and Application,
     The Third Surveying and Mapping Institute of Hunan Province, Changsha 410018, China
*   Correspondence: navydj@163.com

**Abstract:** With the increasing demand for high-precision and difficult-to-obtain geospatial point cloud data copyright protection in military, scientific research, and other fields, research on lossless watermarking is receiving more and more attention. However, most of the current geospatial point cloud data watermarking algorithms embed copyright information by modifying vertex coordinate values, which not only damages the data accuracy and quality but may also cause incalculable losses to data users. To maintain data fidelity and protect its copyright, in this paper, we propose a lossless embedded watermarking algorithm based on vertical stability. First, the watermark information is generated based on the binary encoding of the copyright information and the code of the traceability information. Second, the watermark index is calculated based on the length of the watermark information after compression and the vertical distribution characteristics of the data. Finally, watermark embedding is completed by modifying the relative storage order of the corresponding data according to the index and watermark value. The experimental results show that the proposed algorithm has good invisibility without damaging the data accuracy. In addition, compared with existing algorithms, this method has a higher robustness under operations such as projection transformation, precision perturbation, and vertex deletion of geospatial point cloud data.

**Keywords:** lossless watermarking; geospatial point cloud data; vertical stability; data storage order; robustness

## 1. Introduction

Geographic point cloud data have extremely high military, scientific, social, and economic values and play an important foundational role in multiple fields, such as military operations, scientific research, map drawing, autonomous driving, and smart cities [1–3]. However, with the continuous growth of data sharing demands and increasingly convenient data transmission, the illegal use and infringement of geographic point cloud data occur frequently [4,5]. Data leakage not only damages the interests of the data owners but also poses a threat to national security. Moreover, data sharing can fully tap and realize the value of data. Therefore, effective technical means are urgently needed to protect the copyrights of geographic point cloud data.

As an effective means of protecting data copyrights, digital watermarking technology has been widely applied in copyright protection and leakage tracing of 3-D point cloud data [6,7]. Geographic point cloud data, as a type of 3-D point cloud data, share similar attributes and structural characteristics [8]. Therefore, although research on watermarking algorithms for geographic point cloud data is relatively scarce, the design of such algorithms can refer to the rich achievements of watermarking algorithms for 3-D point cloud data.

Currently, 3-D point cloud watermarking technology primarily embeds watermark information by modifying the coordinate values in the spatial domain [8–14] or frequency domain [15–20] or by modifying the attribute values of the point cloud data, such as the color information [21]. However, these methods inevitably cause damage to the data as they modify the original data's coordinates or attributes. In fields such as navigation and military operations, high precision is required for geographic point cloud data [22–24], and even slight changes in the data coordinates are unacceptable. Therefore, there is an urgent need for watermarking algorithms that do not compromise data accuracy and can replace traditional watermarking algorithms.

Lossless watermarking is a watermarking algorithm that can achieve copyright protection without compromising data accuracy [25–28]. Existing research on lossless watermarking techniques which may be applicable to 3-D point cloud data can be divided into three categories. The first category is reversible watermarking [27,29], which protects the copyright by embedding watermark information in the original data and can restore the data to its original state after extracting the watermark information. For example, algorithm [29] embeds the watermark by modifying the highest frequency coefficient mapped to an integer discrete cosine transform (DCT) domain in the original data and restores the original data based on the inverse process of the watermark embedding after data clustering during the watermark extraction. However, the reversible watermarking approach also terminates data protection when the watermark is extracted. Therefore, this method cannot meet the permanent protection requirements of users for watermarking technology and cannot solve the problem of lossless protection of data throughout the entire process.

The second type is zero-watermarking [10], which generates a watermark using the specific characteristics of the data without modifying the original data. The generated watermark is stored in an intellectual property rights (IPR) repository for future watermark detection. For example, algorithm [10] constructs a watermark image using the attribute value characteristics of the data at the 6th level node after octree partitioning. However, due to the design principles, zero-watermarking methods only construct a watermark without embedding it, which poses the risk of misjudgment in copyright authentication, i.e., the uniqueness verification problem of zero-watermarking algorithms. Additionally, because the watermark is not embedded, different watermark information cannot be extracted from the data using an extraction method, i.e., it cannot achieve traceability.

The third type is the lossless watermarking technique based on storage features [30–32], which was first proposed in the field of lossless watermarking algorithms for vector data. It can embed watermark information without modifying the coordinate values by modifying the storage order of the data according to specific rules, thus achieving lossless embedding of watermark information. For example, algorithm [30] establishes an index based on the angle of the starting point of a line segment and embeds the watermark by reversing the storage order of the nodes within the line segment. Algorithm [31] improves on algorithm [30] by unifying discrete points into line pairs, establishing an index based on the internal angle of the line pairs, and embedding the watermark by reversing the storage order of the line pairs. However, the embedding of watermark information using this type of method is limited by the data type of the carrier, and the method applicable to geographic point cloud data [30] has poor robustness against projection attacks and precision perturbation attacks.

In summary, reversible watermarking methods can restore data during watermark extraction, but they can only achieve one-time data copyright protection. Zero-watermarking methods do not cause any damage to vector data, but they have problems with verification uniqueness and traceability. Lossless watermarking methods based on data storage features can achieve copyright protection and traceability under the condition of lossless precision, but currently, research on such methods is in the early stages, with few cases, and the methods applicable to geographic point cloud data have poor robustness.

To address the aforementioned issues, this paper proposes a lossless embedded and blind watermarking method for geographic point cloud data based on vertical stability.

We establish a watermark index based on the stable vertical attributes of geographic point cloud data, and we correspond the data within different vertical attribute intervals to watermark bits through layering according to the vertical attributes. Then, we modify the relative storage order of the data within each layer in a manner that follows the reverse order of the vertical attributes to achieve lossless embedding of the watermark information. The main contributions of the proposed work are:

1.   Proposal of two feature invariants, the relative size relationship of vertical attributes and the data storage order, for geographic point cloud data.
2.   Proposal of a robustness model of blind and lossless embedded watermarking for geographic point cloud data.

The rest of this paper is organized as follows. Section 2 introduces the design principles of the proposed algorithm. Section 3 presents the algorithm and its implementation. Section 4 presents the experimental results and analysis. Section 5 discusses the findings. Finally, Section 6 presents the conclusions.

## 2. Preliminaries

Compared to other geospatial vector data, the structure of geospatial point cloud data consisting of only three-dimensional discrete points is simpler and has a lower regularity, resulting in fewer invariant features that can be constructed. In the research process, we found that the vertical attribute of the data is independent of the plane coordinates and can remain unchanged during plane geometric transformations such as translation, scaling, rotation, and projection conversion, making it relatively stable. In addition, the relative storage order of the data points is also one of the invariant features of geospatial point cloud data. Therefore, this paper proposes a watermarking scheme based on the vertical attribute and relative storage order of point cloud data. Before introducing the proposed scheme, the following three questions need to be answered: (1) What are the invariant features of geospatial point cloud data? (2) How can these invariant features be used to establish a robust index? (3) How can these invariant features be used to embed watermark information to ensure watermark robustness? The solutions to these three problems are the key to implementing the algorithm proposed in this paper.

### 2.1. Invariant Features of Geographic Point Cloud Data

#### 2.1.1. Vertical Attribute Stability

Geographic point cloud data have three-dimensional attributes and can be divided into planar coordinate attributes and vertical attributes. Therefore, vector point data watermarking algorithms designed based on planar coordinates are applicable to this type of data. However, these methods are not the optimal solution for geographic point cloud data because their direct application ignores the important feature of geographic point cloud data, which is the distance value perpendicular to the horizontal plane [8]. This feature remains unchanged after undergoing processes such as model rotation, translation, and projection transformation in the application of geographic point cloud data because these processes are essentially based on planar coordinate transformations and do not affect the changes in the height positions of each vertex. Additionally, the ground is a tightly ordered system of elevations [33], which not only consists of individual points but also includes the ordered relationships between the elevations of all of the points on the ground; this is the most important essential characteristic of the ground. Figure 1 shows the impact on mountains when the principle of ordered elevations is violated.

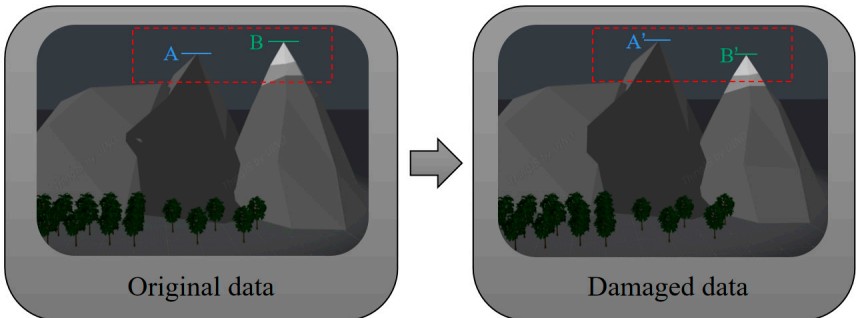

**Figure 1.** Changes in mountain peak vertices.

In Figure 1, it can be seen that after the relative height of the two peaks changes, the main peak of the entire mountain range also changes, and the data model exhibits a significant deviation from the real geographic environment. Therefore, in dealing with the vertical attribute of geographic point cloud data, the high program homogeneity principle [33] should be followed, that is, "high places remain high, and low places remain low". Once this principle is violated, the authenticity of the data will be greatly compromised, and this may even cause incalculable losses to the users. Therefore, the size relationship of the vertical attribute of geographic point cloud data is stable.

2.1.2. Relative Storage Order Stability

As previously mentioned, Zhou [30,32] and Ren [31] achieved lossless watermark embedding in vector geospatial data in their schemes by modifying the relative storage order of the data in a specific way. The relative storage order of the data can remain unchanged after geometric attacks, noise attacks, and projection transformation attacks. Moreover, this property is rarely noticed, so it is almost immune to targeted attacks. Notably, 3-D point cloud data are usually composed of multiple 3-D geographic coordinate records, and there is no explicit requirement for the order of the records or the need to store the topological structure between points. Therefore, it is feasible to embed watermarks by changing the relative storage order of the data. In this study, experiments were conducted to test the performance of the relative storage order of the original part of the data after adding or deleting a 3-D geographic coordinate record (Figure 2).

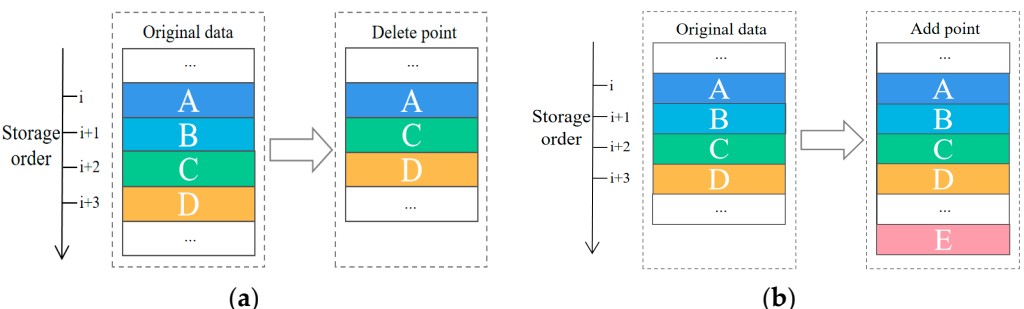

**Figure 2.** The relative storage order of the data after deleting or adding: (**a**) Deleting a data point (**b**) Adding a data point.

Figure 2 shows the deletion and addition of point data, where the storage order of the original point data is A, B, C, and D. As shown in Figure 2a, when point C is deleted, the storage order of the original data becomes A, B, and D. As shown in Figure 2b, when point E is added, the storage order of the original object remains A, B, C, and D. Figure 1 shows that the relative storage order of the original data points remains consistent in every way after deleting and adding point data. Therefore, the relative storage order of the geographic point cloud data is stable.

### 2.2. Index Calculation Based on Vertical Partitioning

In designing the watermark index, first, the geospatial point cloud data are grouped according to the vertical attributes, and then a voting mechanism is used to map multiple groups of data to one set of watermark information, establishing an index relationship between the data and the watermark. Therefore, the specific grouping method of the data directly affects the robustness of the watermark algorithm against cropping attacks. However, the large amount of geospatial point cloud data increases the computational burden of the rational grouping of data. To balance the contradiction between computational efficiency and rational grouping, an equal-distance interval fast grouping method is used. The goal is to maximize data utilization while ensuring that each group of data is evenly distributed.

The specific steps of the index calculation procedure are as follows.

Step 1: To facilitate representation, the vertical attribute values of the data in the formula calculation are denoted as $Z$, and the data points are arranged in ascending order of values. In addition, the points with equal values are considered as one point without distinction.

Step 2: Based on arranging the values in ascending order, the calculation may use a set of different grouping intervals $D = \{d_i | i = 1, 2, 3, \cdots\}$, which can be expressed as follows:

$$d_i = k \times acc \times adj \tag{1}$$

$$d_i \leq (Z_{\max} - Z_{\min}) \tag{2}$$

where $k$ are natural numbers, $acc$ is the minimum precision unit of the data, $Z_{\max}$ is the maximum value, $Z_{\min}$ is the minimum value, and $adj$ is the parameter used for adjusting the grouping interval.

Step 3: A set of grouping intervals $D_1 = \{d1_j | j = 1, 2, 3, \cdots\}$ that can satisfy the premise of watermark embedding is selected and obtained from set $D$. The formula is as follows:

$$num_{d1}^{group} \geq L \tag{3}$$

$$num_{d1}^{p} \geq 2 \tag{4}$$

where $num_{d1}^{group}$ is the number of groups into which the data are grouped according to the interval $d1$, $L$ is the length of the embedded watermark, and $num_{d1}^{p}$ is the number of points in group $P$ when grouping according to interval $d1$.

Step 4: To reduce the subsequent computational complexity, variance is used to select a set of $t$ intervals $D_2 = \{d2_k | k = 1, 2, 3, \cdots, t\}$ that can most evenly divide the watermark embedding interval. The formula is as follows:

$$\delta_{d1} = \sum_{p=1}^{p=n} \left(num_{d1_j}^{p} - \overline{num_{d1_j}}\right)^2 \tag{5}$$

where $\delta_{d1_j}$ is the uniformity of the data grouping using interval $d1_j$, and the smaller the value is, the more uniform the grouping is. $\overline{num_{d1_j}}$ is the average number of points within each group when grouped using interval $d1_j$.

Step 5: The grouping method that can make the best use of the data is selected from the set of interval ranges $D_2$ using the following formula:

$$num_{d2_k}^{surplus} = num_{pt} - num_{d2_k}^{cycle} \tag{6}$$

where $num_{d2_k}^{surplus}$ is the total number of remaining data points after grouping using interval ranges $d2_k$, and only using these remaining data points cannot be fully embedded with the watermark information once. $num_{pt}$ is the total number of data points. $d2_k$ is the number of

data points used to embed the watermark information after grouping using interval ranges $d2_k$. The interval range $num_{d2_k}^{surplus}$ used when the minimum value is taken is denoted as $d_{perf}$, which is the actual interval range used to embed the watermark.

### 2.3. Watermark Embedding Rules Based on Storage Direction

According to the previous analysis, geospatial point cloud data have a stable relative storage order, and even if data are deleted or added, the relative storage order of the original data does not change. Therefore, based on the grouping of the data points in the previous section, the relative storage order of each group of data is changed based on the size relationship of the vertical attribute Z to achieve watermark embedding. The specific method is as follows: when the watermark bit is 0, the relative storage order of each group of indexed original data is adjusted from small to large according to the Z value (Figure 3a). When the watermark bit is 1, the relative storage order of each group of indexed original data is adjusted from large to small according to the Z value (Figure 3b).

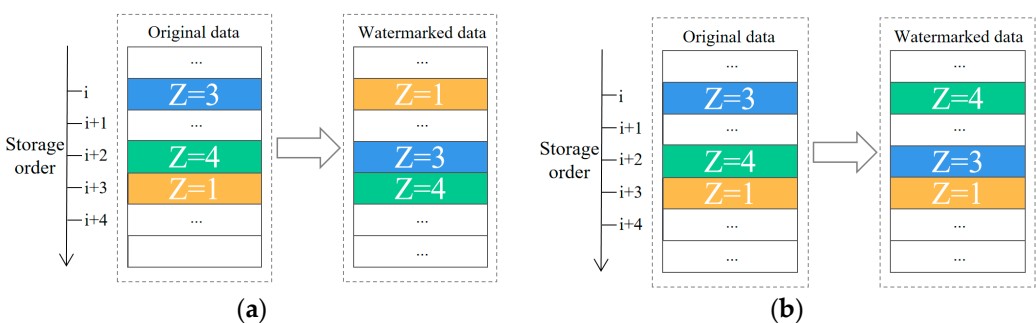

(a)　　　　　　　　　　　　　　　　　　　　　　(b)

**Figure 3.** Demonstration of watermark embedding: (**a**) When the watermark bit is 0; (**b**) When the watermark bit is 1.

### 2.4. Generation of Watermark

As can be seen in the previous two sections, the algorithm proposed in this article is based on the vertical attributes of the data to establish an index and embed the watermark by modifying the storage order of the relative data points. It has robustness and invariance under geometric attacks, precision perturbation attacks, and projection transformation attacks. However, when facing data-cropping attacks, the algorithm's robustness often needs to be enhanced through redundant embedding based on a voting mechanism. To improve the voting rate of each watermark bit, the algorithm uses an eight-bit binary code to describe the copyright information of a letter. For example, the uppercase letter A is described using 01000001, and the lowercase letter a is described using 01100001.

In addition, the legal flow of data is often directional, and the full information expression of the unit names involved in the flow process would reduce the voting rate of each watermark bit. Therefore, in this article, binary code is used to complete the expression of multi-level tracking objects; that is, a binary number of length $m_1$ is used to represent all of the unit objects involved in the first level of tracking, and a binary code of length $m_2$ is used to represent all of the unit objects involved in the second level of tracking. The nth level of tracking is represented by a code of length $m_n$.

## 3. Methodology

### 3.1. Basic Principle

The purpose of this article is to design a robust, lossless watermarking algorithm for geographic point cloud data. To address the three issues mentioned in Section 2, the following aspects are mentioned in the design of the proposed watermarking algorithm. (1) The size relationship and relative storage order of the vertical attributes of geographic point cloud data are stable, and the algorithm is designed based on this invariant feature. (2) To solve the watermark synchronization problem, an indexing scheme based on vertical

attribute grouping is designed. (3) The watermark bit 1 or 0 is represented in ascending or descending order of the vertical attributes, and based on this rule, the relative storage order of each group of data is adjusted according to the watermark index to embed the watermark. In addition, to improve the voting rate of each watermark bit under the voting mechanism, a simplified watermark information generation scheme is proposed.

In this section, we provide a complete description of the proposed watermarking scheme for geographic point cloud data based on vertical stability. The scheme consists of two steps: watermark embedding and watermark extraction. The following sections provide a detailed description of each procedure.

### 3.2. Watermark-Embedding Process

First, the algorithm compresses the watermark information, which is then reasonably grouped according to the vertical properties of the data to complete the watermark index calculation. Finally, the relative storage order of the data is adjusted to complete watermark embedding. The proposed embedding process is shown in Figure 4.

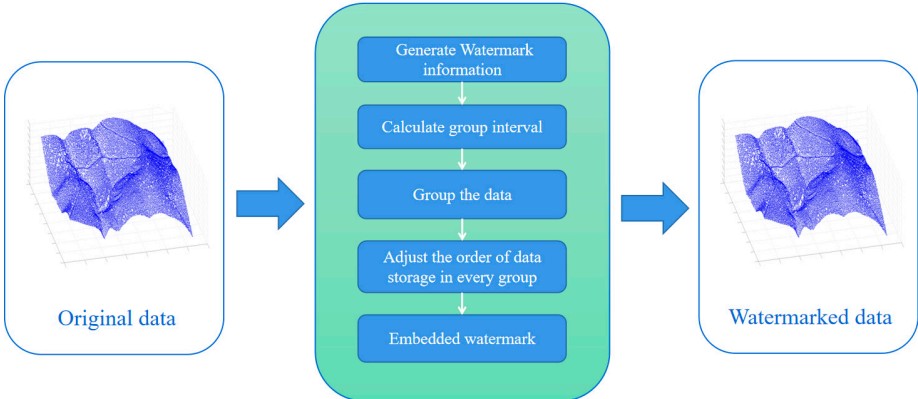

**Figure 4.** Watermark-embedding process.

The specific steps of the watermark-embedding process are as follows.

Step 1: Generate the watermark *code* in the manner described in Section 2.4 and calculate the group spacing $d_{perf}$ based on the length.

Step 2: Arrange the data separately in ascending order according to $Z$, and group them according to the group spacing $d_{perf}$.

Step 3: Arrange the $Z$ values in descending $Order_{max}$ and in ascending $Order_{min}$. Modify the relative storage order of the corresponding data points in each group according to the watermark sequence. The formula is as follows:

$$group_{pt}(R \times ii) = \begin{cases} Order_{max}, & code(ii) = 0 \\ Order_{min}, & code(ii) = 1 \end{cases} \tag{7}$$

where $group_{pt}(R \times ii)$ represents the relative storage order of the corresponding data points in group $R \times ii$ after data grouping, $code(ii)$ represents the value of the bit in the watermark code sequence, and $R$ represents the embedded ordinal number.

Step 4: Record the group interval $d_{perf}$ and the starting value $z_{min}$ of each group as the key for the storage.

### 3.3. Watermark-Extraction Process

The proposed watermark algorithm is a blind watermark algorithm, and the extraction process does not require the involvement of the original data. The watermark-extraction process is similar to the embedding process, except that the integrity of the relative storage order within each group of the original data may be compromised by vertex addition. Therefore, the watermark detection value is determined by the relative arrangement direction of the longest $Z$ value size in the group. Figure 5 illustrates the process of determining

the longest sequence order after adding data point interference. After the data embedded with the watermark are subjected to vertex addition attacks, the *Z* value arrangement based on the storage order in a certain group changes from 1, 2, 4 to 1, 2, 4, 3. The longest sequence order from small to large is 1, 2, 4, and the longest sequence order from large to small is 4, 3. The longest sequence order is still arranged from small to large, which is 1, 2, 4, and its watermark detection value is 1.

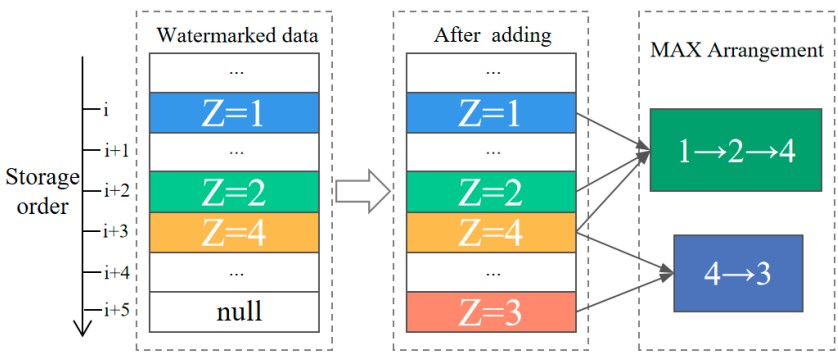

**Figure 5.** Process of determining the longest sequence order.

The proposed extraction process is shown in Figure 6.

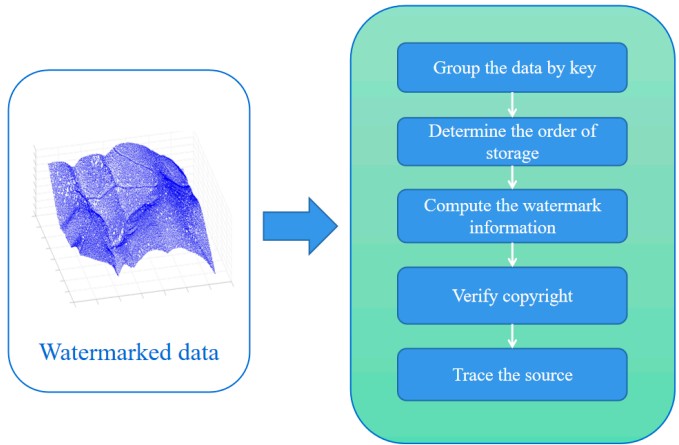

**Figure 6.** Watermark-extraction process.

The specific steps of the extraction process are as follows.

Step 1: Group the data based on the key and the group spacing $d_{perf}$ starting from $z_{\min}$ according to the value *Z* while maintaining the relative storage order of the data.

Step 2: The longest sequence of values *Z* within each group is denoted as $arrangement^{longest}$ when sorted in descending order $arrangement^{longest}_{\max}$ and denoted as $arrangement^{longest}_{\min}$ when sorted in ascending order. Accordingly, calculate the corresponding watermark bit decision value $judge_{m,n}$ for each group using the following equation:

$$judge_{m,n} = \begin{cases} -1, arrangement^{longest} = arrangement^{longest}_{\max} \\ 1, arrangement^{longest} = arrangement^{longest}_{\min} \end{cases} \tag{8}$$

where *m* is the position of the watermark and *n* represents the order of the watermark embedding loop.

Step 3: Calculate the watermark encoding based on the decision value $judge_{m,n}$. The formula is

$$code(m) = \begin{cases} 0, \sum_n judge_{m,n} < 0 \\ 1, \sum_n judge_{m,n} \geq 0 \end{cases} \tag{9}$$

　　　　Step 4: Decode the watermark encoding to obtain the copyright information and traceability information carried within it.

### 4. Experimentation and Analysis

*4.1. Experimental Preparation*

4.1.1. Experimental Data and Watermark Information

　　　　In this experiment, two different types of geographic point cloud data, namely, underwater terrain and land terrain data, in .txt format, were used as the experimental data. As shown in Figure 7, the relevant parameters of the data are shown in Table 1.

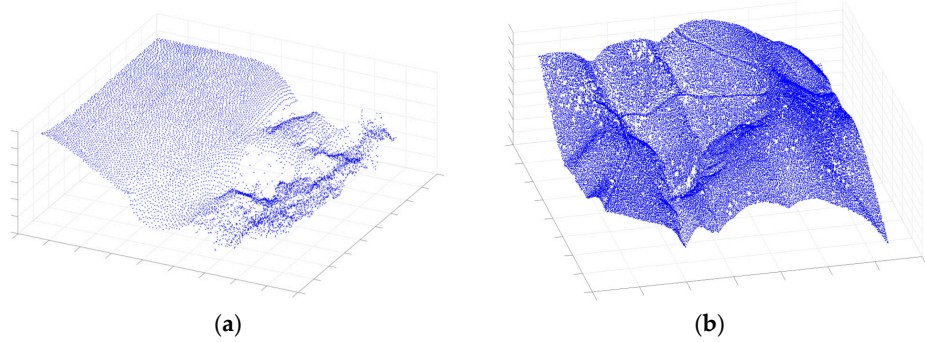

|(**a**)|(**b**)|

**Figure 7.** Experimental data: (**a**) Seafloor topography data. (**b**) Land topography data.

**Table 1.** Basic information about the experimental data.

| Data Format | Geographic Types | Number of Vertices | Acquisition Method | Planar Coordinate Accuracy | Data Size |
|---|---|---|---|---|---|
| TXT | Seafloor topography | 9814 | Multi-beam measurement | Accurate to six decimal places after the decimal point | 384 KB |
| TXT | Land topography | 49,943 | Laser scanning | Accurate to six decimal places after the decimal point | 2878 KB |

　　　　The copyright information "Watermark" is composed of 72 binary codes. The traceability information simulates four levels of tracking, with each level containing 10 binary codes that can describe 1028 different objects for a total length of 40. The total length of the watermark information is 112.

4.1.2. Evaluation Criteria

　　　　When detecting watermark information for suspicious or attacked data, it is necessary to compare the extracted watermark information with the copyright information registered with the copyright registration center for similarity and then to evaluate the watermark similarity. The normalized correlation coefficient (*NC*) is used to evaluate the robustness of the watermark algorithm, and in this article, the threshold is set to 0.8, which is based on the empirical test and analysis of the algorithm's performance in various test cases and provides a good balance between the ability to extract the watermark accurately and the robustness of the algorithm to random vertex deletion or other distortion [34]. If the value

is greater than or equal to the threshold, the watermark can be extracted; otherwise, the watermark cannot be extracted. The calculation process is as follows:

$$NC = \frac{\sum\limits_{i=1}^{i=M} \sum\limits_{j=1}^{j=N} XNOR(W(i,j), W'(i,j))}{M \times N} \tag{10}$$

where $M \times N$ is the size of the watermark image, $W(i,j)$ is the original watermark, $W'(i,j)$ is the extracted watermark, and $NC$ is calculated with a precision of two decimal places and rounded down.

The change rate ($CR$) is used to evaluate the losslessness of a watermarking algorithm, defined as the ratio of the coordinate value changes of the watermark data vertices relative to the original data. When the $CR$ is equal to 1, the watermark has no impact on the coordinate values of the data, achieving precision losslessness. The formula for calculating the $CR$ is as follows:

$$CR = \frac{stats\left(\sqrt{(x_i' - x_i)^2 + (y_i' - y_i)^2 + (z_i' - z_i)^2}\right)}{N_{vertices}} \tag{11}$$

where $N_{vertices}$ is the total number of coordinate points in the data; $stats()$ is used to count the number of changes in the x, y, and z coordinates of the coordinate points; and $(x_i, y_i, z_i)$ and $(x_i', y_i', z_i')$ are the coordinate points of the original data and the watermarked data, respectively.

The root mean square error ($RMSE$) is used to evaluate the imperceptibility of the watermarking algorithm. When the $RMSE$ is equal to zero, the watermark is completely invisible, achieving imperceptibility. The calculation formula is as follows:

$$RMSE = \sqrt{\frac{\sum\limits_{i=1}^{i=n} (x_i' - x_i)^2 + \sum\limits_{i=1}^{i=n} (y_i' - y_i)^2 + \sum\limits_{i=1}^{i=n} (z_i' - z_i)^2}{n}} \tag{12}$$

where $n$ is the total number of coordinate points in the data, and $(x_i, y_i, z_i)$ and $(x_i', y_i', z_i')$ are the coordinate points of the original data and the watermarked data, respectively.

### 4.2. Analysis of Losslessness and Invisibility

To evaluate the performance of the proposed method, in this section, we conduct experiments with Liu's algorithm [11] based on the spatial domain and Lepus's algorithm [15] based on the frequency domain. We use Equations (11) and (12) to calculate the $CR$ and $RMSE$ values, respectively, for embedding watermarks of two different geographical types in the geographic point cloud data using our algorithm and two comparative algorithms, and we take the maximum value as the calculation result (Table 2). The results show that the $CR$ and $RMSE$ values of the proposed are always 0; however, none of the values of the two are 0. This is because the watermark embedding carrier of the proposed algorithm is the relative storage order of the data rather than the coordinate attribute value, thus resulting in no damage to the data attributes. Therefore, our algorithm outperforms the comparison algorithm in terms of losslessness and invisibility, which is consistent with the theoretical analysis and experimental results.

**Table 2.** Experimental results for losslessness and invisibility.

| Experimental Data | CR | | | RMSE | | |
|---|---|---|---|---|---|---|
| | Proposed | Liu [11] | Lipuš [15] | Proposed | Liu [11] | Lipuš [15] |
| data (a) | 0 | 0.76 | 0.69 | 0 | 0.0043 | 0.00071 |
| data (b) | 0 | 0.73 | 0.64 | 0 | 0.0039 | 0.00065 |

### 4.3. Robustness Analysis

This section presents a robustness analysis of the watermarking algorithm proposed for testing. Fully considering the possible operational methods in the data application scenarios as well as the approaches that infringers may take to destroy watermarks, we conduct experiments on RST (rotation, scaling, and translation) attacks, precision perturbation attacks, projection transformation attacks, and random deletion attacks. The watermark proposed in this paper and Ren's watermark (a lossless embedded watermarking algorithm similar to the proposed algorithm) [25] are embedded in two different geographical types of point cloud data, and the values extracted from both types of data after being attacked are calculated. The larger the value is, the better the robustness performance is.

#### 4.3.1. The Robustness of RST

Rotation, scaling, and translation are the most common operations for geographic data and are also the most common types of geometric attacks on geographic point cloud watermarks. To evaluate the robustness of the algorithm against RST attacks, experiments are conducted on different degrees of RST attacks on the algorithm proposed in this paper and Ren's algorithm. The rotation experiment rotates the data from $30°$ to $330°$ at intervals of $30°$. The translation experiment is designed with intervals of 20% of the maximum range of the vector data border length, and the data are translated from 10% to 210% of the original data ratio. The scaling experiment is designed with intervals of 20%, and the data are scaled from 10% to 210% of the original data ratio. The *NC* value calculation results after the RST attack are shown in Figure 8. The results show that the *NC* value of the watermark results extracted by the watermark algorithm proposed in this paper and the comparison algorithm is always 1 under different intensities of RST attacks. This is because both algorithms are based on feature invariants that can effectively resist RST attacks. For example, the method proposed uses the vertical stability of point cloud data, and Ren's method uses the stability of the angle between lines composed of points. Therefore, the watermark algorithm proposed in this paper based on vertical stability has strong robustness and invariance against geometric attacks, and the experimental results are consistent with the theoretical analysis.

#### 4.3.2. The Robustness of Precision Perturbation

In the case of facing copyright infringers who disrupt watermarks by tampering with data coordinates, precision perturbation experiments can be used to verify the robustness of watermark algorithms. In this section, the precision perturbation attack refers to adding random errors of 0–9 to different digits of the x and y coordinates of the data. The minimum intensity attack is to add an error to the last digit, and the maximum intensity attack is to add an error to the eighth digit from the end, with a step size of 1. The NC value calculation results after the precision perturbation attack are shown in Figure 9. The calculation results show that the watermark algorithm based on the vertical stability proposed in this paper always has an NC value of 1 as the randomly added error position moves forward, while Ren's algorithm has an NC value of 1 at the beginning, but the NC value starts to decrease when the error position moves to the fourth digit from the end. Starting from the seventh digit from the end, the NC value of both datasets is below the threshold. This is because the watermark algorithms assessed in this paper use the *Z* value of the data to establish the watermark and the index of the data, so the precision perturbation of the plane coordinates does not affect the extraction of the watermark. However, Ren's algorithm establishes the watermark and the index of the data based on the angle between the lines, and larger random errors change the angle between the lines, thereby destroying the index of the watermark. Therefore, the watermark algorithm based on the vertical stability proposed in this paper is significantly superior to Ren's algorithm when facing precision perturbation attacks, and it has strong robustness and invariance to precision perturbation attacks. The experimental results are consistent with the theoretical analysis.

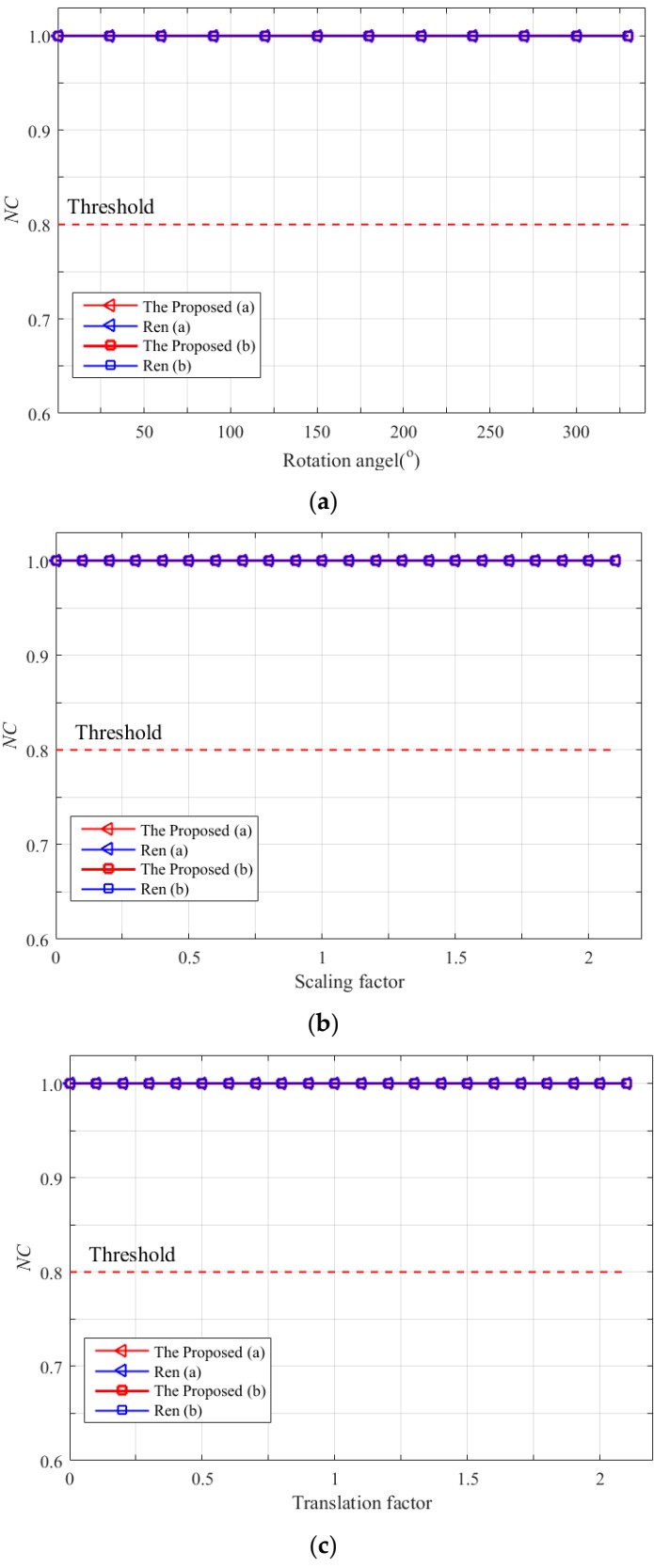

**Figure 8.** RST Attack result: (**a**) Rotation attack. (**b**) Scaling attack. (**c**) Translation attack.

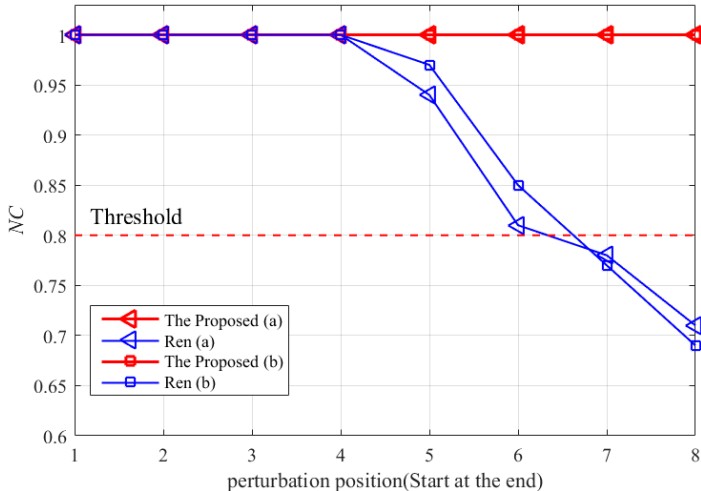

**Figure 9.** Precision perturbation attack results.

### 4.3.3. The Robustness of Projection Transformation

A projection transformation attack is a common attack method that targets geospatial data, attempting to compromise the extractability of watermarks by applying projection transformations to the data. For example, attackers may attempt to first use projection transformations to disrupt the watermark and then use reverse projection transformations to restore the original data in order to evade checking and tracking of the watermark. To evaluate the algorithm's robustness to a projection transformation attack, four types of projections were used in this experiment, namely, equal-area projection, equal-angle projection, equidistant projection, and compromise projection. The first three types ensure that the true area, angle, and distance values of the data are not affected by the projection, while the compromise projection ensures that all of the factors are kept as consistent as possible within a small view rather than a large view. As shown in Table 3, each of the four types of projections contains three specific projection methods, for a total of twelve projection methods, which demonstrates the distortion of the geographic point cloud data in as many ways as possible and more effectively validates the robustness of the proposed algorithm.

**Table 3.** Map projections.

| Projection Type | Projection Name | Short Name |
|---|---|---|
| Equal area projection | Equal-Area Cylindrical Projection | Eqacylin |
| | Gall Orthographic Projection | Gortho |
| | Lambert Azimuthal Equal-Area Projection | Eqaazim |
| Conformal projection | Mercator Projection | Mercator |
| | Lambert Conformal Conic Projection | Lambert |
| | Stereographic Projection | Stereo |
| Equidistant projection | Equidistant Azimuthal Projection | Eqdazim |
| | Equidistant Cylindrical Projection | Eqdcylin |
| | Equidistant Conic Projection | Eqdconic |
| Compromise projection | Robinson Projection | Robinson |
| | Winkel I Projection | Winkel |
| | Aitoff Projection | Aitoff |

For the 12 projection methods listed in Table 3, we compared our proposed algorithm with Ren's algorithm. Tables 4–7 show the *NC* value calculation results for the different watermark schemes under four types (12 subtypes) of projections and for two experimental datasets.

**Table 4.** The results after equal area projection transformation.

| Projection Number | Projection Name | Experimental Data | NC | |
|---|---|---|---|---|
| | | | Proposed | Ren [25] |
| 1 | Eqacylin | data (a) | 1 | 0.81 |
| | | data (b) | 1 | 0.82 |
| 2 | Gortho | data (a) | 1 | 0.77 |
| | | data (b) | 1 | 0.79 |
| 3 | Eqaazim | data (a) | 1 | 0.95 |
| | | data (b) | 1 | 0.94 |

**Table 5.** The results after conformal projection transformation.

| Projection Number | Projection Name | Experimental Data | NC | |
|---|---|---|---|---|
| | | | Proposed | Ren [25] |
| 4 | Mercator | data (a) | 1 | 1 |
| | | data (b) | 1 | 1 |
| 5 | Lambert | data (a) | 1 | 1 |
| | | data (b) | 1 | 1 |
| 6 | Stereo | data (a) | 1 | 1 |
| | | data (b) | 1 | 1 |

**Table 6.** The results after equidistant projection transformation.

| Projection Number | Projection Name | Experimental Data | NC | |
|---|---|---|---|---|
| | | | Proposed | Ren [25] |
| 7 | Eqdazim | data (a) | 1 | 0.89 |
| | | data (b) | 1 | 0.90 |
| 8 | Eqdcylin | data (a) | 1 | 0.93 |
| | | data (b) | 1 | 0.94 |
| 9 | Eqdconic | data (a) | 1 | 0.82 |
| | | data (b) | 1 | 0.81 |

**Table 7.** The results after compromise projection transformation.

| Projection Number | Projection Name | Experimental Data | NC | |
|---|---|---|---|---|
| | | | Proposed | Ren [25] |
| 10 | Robinson | data (a) | 1 | 0.85 |
| | | data (b) | 1 | 0.84 |
| 11 | Winkel I | data (a) | 1 | 0.82 |
| | | data (b) | 1 | 0.82 |
| 12 | Aitoff | data (a) | 1 | 0.83 |
| | | data (b) | 1 | 0.82 |

It can be clearly seen in Tables 4–7 that the *NC* value of the proposed method is 1 under the 12 projection attacks and for the 2 experimental datasets. However, the *NC* value

of Ren's algorithm changes with the map projection method. This is because different projection methods can indeed affect the representation and size of objects on a plane, but height (vertical property) refers to the vertical distance between an object and a reference plane (usually a horizontal plane), and changing the projection method will not affect the height. However, Ren's method is based on the angle between lines to establish a watermark and the index between the data, and the angle changes under other projection methods, except for equal-angle projection. Therefore, the *NC* value is only 1 under the three types of equal-angle projection, which is lower than that of the algorithm proposed in this paper under the other types of projection, and even lower than the threshold value under the Gortho projection transformation attack. In addition, Figure 10 shows that the proposed method maintains an *NC* value of 1 under the 4 types (12 subtypes) of projections and for the 2 datasets. Therefore, the watermark algorithm based on vertical stability proposed in this paper is significantly better than Ren's algorithm under projection attacks, and it has strong robustness and invariance to projection attacks. The experimental results are consistent with the theoretical analysis.

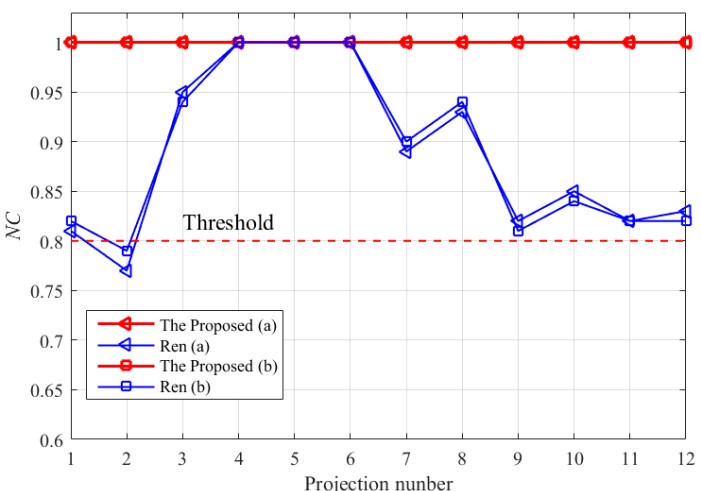

**Figure 10.** The results of the projection attacks.

### 4.3.4. The Robustness of Random Deletion

A random deletion attack is a common attack method where attackers randomly remove some information from the watermark to simulate data corruption or tampering that may occur in real-world applications. By conducting such attack experiments, the performance of watermark algorithms can be evaluated when facing data corruption or tampering in order to determine their ability and reliability in countering attacks. In this section, a random deletion attack refers to the random deletion of some vertices from point cloud data, and it is one of the most important attacks that watermark algorithms need to resist. The minimum number of deleted points is 5% of the total data, and the maximum is 50%, with a step size of 5%. The remaining parts of the two types of data under deletion ratios of 25% and 50% are shown in Figures 11 and 12, respectively. The *NC* value calculation results are shown in Figure 13, which shows that both algorithms are affected by random vertex deletion attacks, but the algorithm proposed in this paper has a better ability to resist vertex deletion attacks than Ren's algorithm. This is because random deletion attacks do not destroy the watermark index established based on the vertical attribute interval of the data, i.e., the algorithm proposed in this paper, and the watermark extraction based on the voting mechanism can effectively resist data deletion attacks. Ren's watermark index is established based on the normal storage order of data and is easily disrupted by random deletion, resulting in some elasticity in its *NC* calculation results. The theoretical analysis and experimental results are consistent, and the algorithm proposed in this paper has better robustness against random deletion attacks.

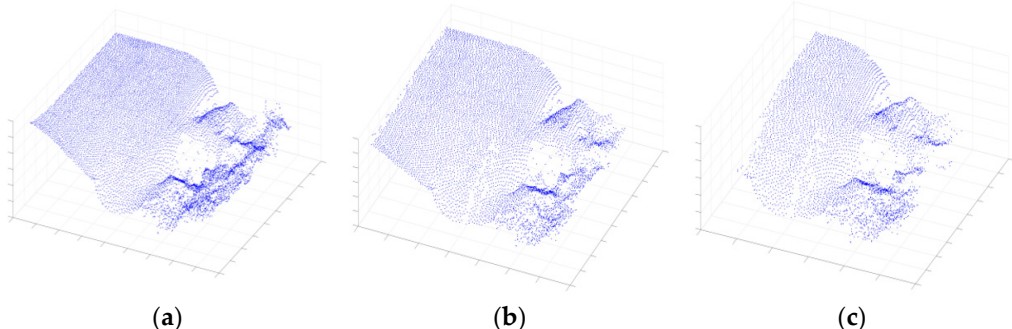

**Figure 11.** Random deletion experiments: (**a**) Original seafloor topography data. (**b**) A 25% deletion result. (**c**) A 50% deletion result.

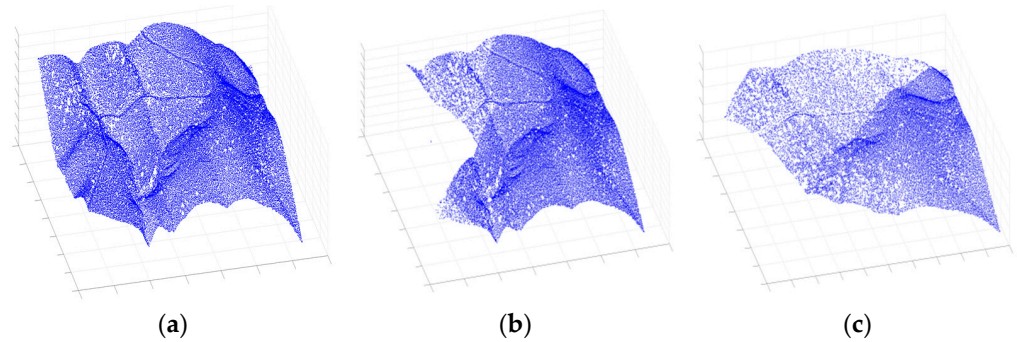

**Figure 12.** Random deletion experiments: (**a**) Original land topography data. (**b**) A 25% deletion result. (**c**) A 50% deletion result.

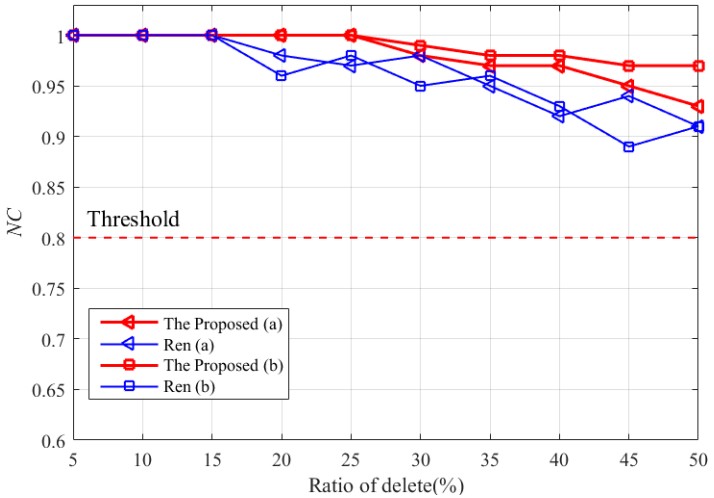

**Figure 13.** The results of random deletion attack.

## 5. Discussion

The algorithm proposed in this paper supplements the existing lossless watermarking algorithm for geographic point cloud data to a certain extent. This has been demonstrated through the analysis of the algorithm's principle and experimental verification. Next, we discuss the algorithm from multiple perspectives, including its applicability and the improvement direction of the data-indexing method under vertical attribute attacks.

### 5.1. Discussion on the Applicability of the Algorithm to Small Datasets

During the experiments, we found that the algorithm proposed in this paper not only solves the problems of copyright verification and the traceability of geographic point cloud data that existing algorithms cannot achieve when the data are completely lossless, but it is also applicable to small geographic point cloud datasets. In this section, we select the water depth points from the vector electronic navigation chart with a scale of 1:10,000 and a chart number of C1613182 as the experimental data to verify and discuss the applicability of the algorithm proposed in this paper. As shown in Figure 14, there are a total of 1199 water depth points in the dataset.

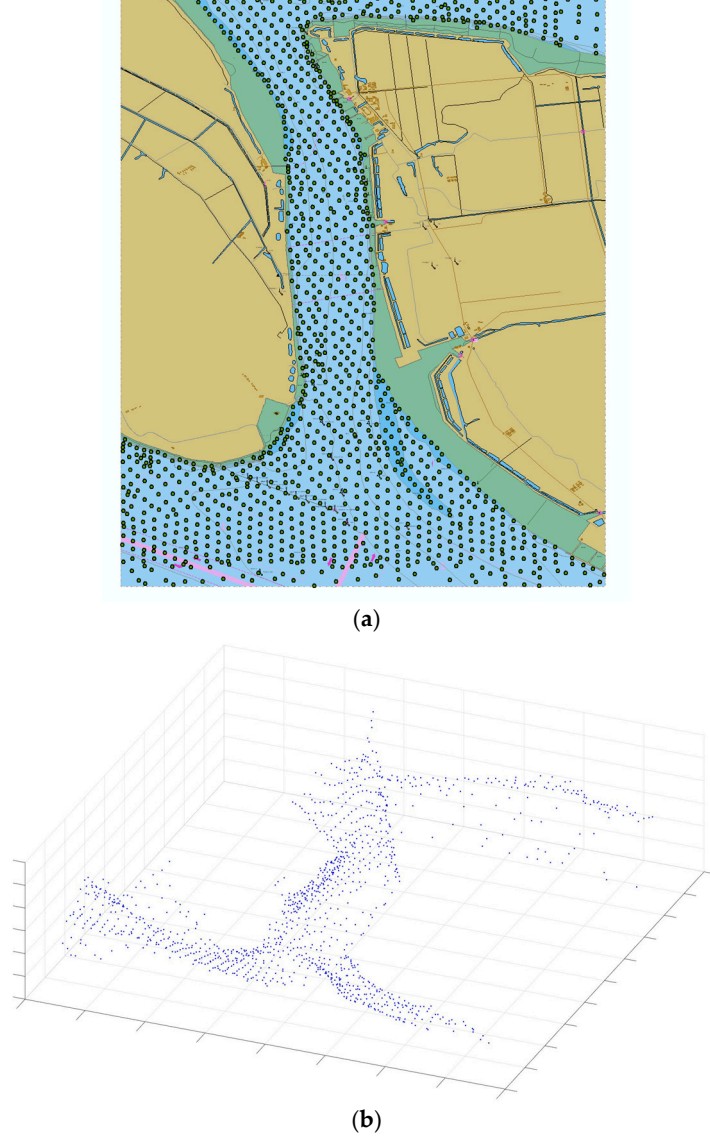

(**a**)

(**b**)

**Figure 14.** Experimental data: (**a**) ENC data (**b**) Three-dimensional illustration of water depth points.

The *NC* value calculation results for the watermark information extracted in the different types of attack experiments are presented in Table 8, which shows that even if the random deletion strength reaches 50%, the *NC* values extracted from the two types of data are still higher than the threshold. The experimental verification shows that the proposed algorithm can be effectively applied to small geographic point cloud datasets, and the lossless watermark algorithm for exploring geographic point cloud data with similar data structures has a certain value and can provide a certain degree of reference for subsequent research.

**Table 8.** Algorithm applicability results.

| | | NC | | |
|---|---|---|---|---|
| Rotation by 120° | Translation by 50% Scaling | by 0.5 12 Types of | Projection Transformations | 50% Deletion |
| 1.00 | 1.00 | 1.00 | 1.00 | 0.89 |

*5.2. Discussion of Different Data-Grouping Methods during Watermark Embedding*

The core process of the proposed algorithm is to group the data based on their vertical attributes. In Section 2.2, a precise calculation-based equidistant grouping method is presented. This grouping method aims to achieve uniform grouping and improve the robustness of the algorithm when applied to small datasets, ensuring the upper limit of the algorithm. For larger datasets, when the watermark length is fixed, each bit can be determined by more data points, resulting in better resistance to deletion attacks. However, as the data volume increases, the computational complexity of precise grouping also increases in a non-linear manner. To address this issue, the watermark-embedding process can be optimized by increasing the search interval of the grouping distance and dividing the data for multiple grouping calculations. To verify the effectiveness of the two aforementioned methods, we selected a large-scale geospatial point cloud dataset (Figure 15) with 9,635,451 data points as the experimental data to conduct experiments on times of the optimized vertical partition calculation and watermark generation. The experiment in this study was conducted using a laptop computer equipped with an Intel Core i7 processor and 16 GB of RAM. The computer ran on the Windows 11 operating system, and the algorithms were implemented and performance evaluated using the MATLAB programming language. Basic information about the geographic point cloud data is shown in Table 9.

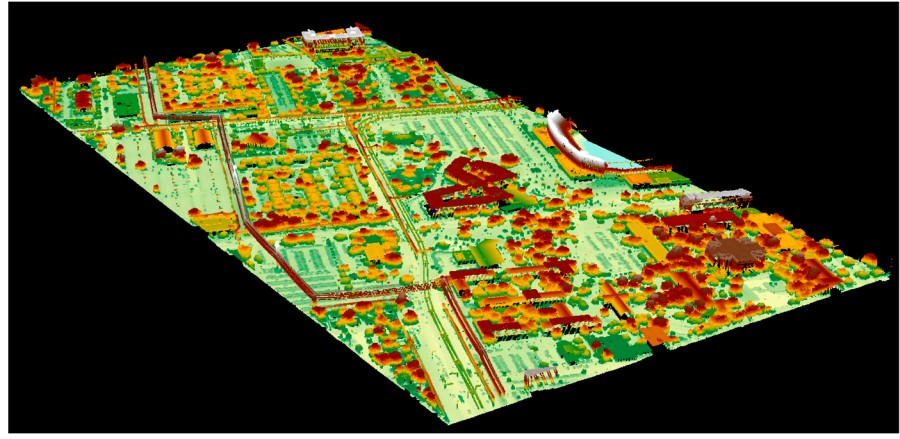

**Figure 15.** Experimental data: Large-scale geospatial point cloud dataset.

**Table 9.** Basic information about the experimental data.

| Data Format | Geographic Types | Number of Vertices | Acquisition Method | Vertical Coordinate Accuracy | Data Size |
|---|---|---|---|---|---|
| TXT | Residential area | 9,365,452 | Laser scanning | Accurate to two decimal places after the decimal point | 259,259 KB |

The time that the proposed algorithm spends on calculating the vertical partitioning and finishing embedding the watermark using the two time-optimization methods simultaneously is shown in Tables 10 and 11. Based on Tables 10 and 11, it can be concluded that both of the two proposed time optimization methods in this paper can effectively reduce the computation time for grouping and the overall watermark generation time. Additionally,

the computation time for grouping consistently accounts for a significant proportion of the overall watermark generation time before and after optimization, indicating that there is still room for improvement in optimizing this aspect.

**Table 10.** Time spent on calculating the vertical partitioning (seconds).

| Increase the Search Interval of Grouping by 1 Time | | | Increase the Search Interval of Grouping by 2 Times | | | Increase the Search Interval of Grouping by 5 Times | | |
|---|---|---|---|---|---|---|---|---|
| Divide the Data into 1 Group | Divide the data into 5 Groups | Divide the Data into 10 Groups | Divide the Data into 1 Group | Divide the Data into 5 Groups | Divide the Data into 10 Groups | Divide the Data into 1 Group | Divide the Data into 5 Groups | Divide the Data into 10 Groups |
| 2068.928 | 1401.652 | 1017.431 | 1087.027 | 758.752 | 641.129 | 487.568 | 389.375 | 317.254 |

**Table 11.** Time spent on generating the watermark (seconds).

| Increase the Search Interval of Grouping by 1 Time | | | Increase the Search Interval of Grouping by 2 Times | | | Increase the Search Interval of Grouping by 5 Times | | |
|---|---|---|---|---|---|---|---|---|
| Divide the Data into 1 Group | Divide the Data into 5 Groups | Divide the Data into 10 Groups | Divide the Data into 1 Group | Divide the Data into 5 Groups | Divide the Data into 10 Groups | Divide the Data into 1 Group | Divide the Data into 5 Groups | Divide the Data into 10 Groups |
| 2075.356 | 1432.953 | 1079.548 | 1093.631 | 791.641 | 702.417 | 494.425 | 423.662 | 382.473 |

Another grouping method that can ensure the robustness of the watermark algorithm is equal point grouping, and it also has the advantage of fast computation speed. However, this method requires recording the grouping basis of each group of data, especially when dealing with large amounts of data, such as geographic point cloud data obtained from laser scanning. This may require a significant amount of storage space, posing a huge burden on key storage. In addition, the feature points of geographic point cloud data, such as the maximum and minimum points, are more important than ordinary points in practical applications, and infringers have to consider selectively retaining them when maliciously deleting data, making the feature points more stable. Therefore, the method of data grouping can affect the performance and applicability of the algorithm. Users of the algorithm can modify the method of data grouping to improve its performance based on the specific characteristics of the data.

## 6. Conclusions

This paper proposes a new lossless embedded watermarking algorithm for geographic point cloud data, which is based on the vertical stability of the data and establishes an index between the watermark and the data. Then, the watermark is embedded by adjusting the relative storage order of the corresponding data. Additionally, the watermark extraction can be completed with just a key, making the proposed blind algorithm convenient for practical applications. The experiments and results show that the proposed algorithm has better robustness than existing storage-feature-based lossless watermarking algorithms. The algorithm complements existing point-data-based lossless algorithms to a certain extent and is also applicable to small datasets and 3-D point cloud model data with similar structures. Furthermore, the algorithm could be significant in exploring lossless watermarking algorithms for similar data and provide a reference for related research. However, the algorithm does not distinguish the importance of vertices in different geographic locations. When infringers modify this attribute artificially, there is still room for improvement regarding how to better utilize geographic feature points to improve robustness, which will be the main focus of our future research.

**Author Contributions:** Conceptualization, Mingyang Zhang, Jian Dong, and Na Ren; methodology, Mingyang Zhang, Jian Dong, and Na Ren; validation, Mingyang Zhang, Jian Dong, and Na Ren formal analysis, Mingyang Zhang and Na Ren; writing—original draft preparation, Mingyang Zhang; writing—review and editing, Mingyang Zhang, Na Ren, and Shuitao Guo; visualization, Mingyang Zhang and Jian Dong; supervision, Jian Dong and Na Ren; funding acquisition, Jian Dong and Na Ren. All authors have read and agreed to the published version of the manuscript.

**Funding:** This work was supported by the National Natural Science Foundation of China under Grant 42071439, the Natural Science Foundation of China under Grant 42071362, and the Research Foundation of the Department of Natural Resources of Hunan Province under Grant 2022-03.

**Data Availability Statement:** The data presented in this study are available on request from the corresponding author. The data are not publicly available due to military sensitivity.

**Conflicts of Interest:** The authors declare no conflict of interest.

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
