# Peer review of "Lossless Watermarking Algorithm for Geographic Point Cloud Data Based on Vertical Stability"

_ijgi, doi:10.3390/ijgi12070294_

Round 1
Reviewer 1 Report
"Lossless Watermarking Algorithm for Geographic Point Cloud Data Based on Vertical Stability" titled manuscript propose a lossless watermarking algorithm based on vertical stability in geographic data. Here is the comments:
1. In watermarking, our aim is to find robust, secure, resist and high data capacity based algorithm for all type data (image, video, text, software). It should not only be for geo data.
2. In the literature, there are lots of algorithm in spatial domain and frequency domain which are very successful for any type of data. In the manuscript, authors did not mention and compare with this algorithms.
3. Blind, semi-blind and non-blind algorithms are also not given in the paper. If we use watermark, original data or only secret key how proposed algorithm results?
4. Attacks. No possible attack is given in the paper. This one is one of the important measurement for the success.
5. Reference list is not current.
Reviewer 2 Report
The proposed manuscript presents a novel lossless watermarking algorithm specifically designed for geographic point cloud data. The algorithm uses the concept of vertical stability to establish a relationship between the watermark and the data. By adjusting the storage order of the corresponding data, the watermark is embedded. The algorithm can serve as a complement to existing lossless algorithms for point data and can be applied to small 3D point cloud datasets.
Below are some specific annotations to improve clarity and determinism:
- It is necessary to explain what the difference is between watermarking techniques for 3D point cloud data and general vector data. The classifications used in the paper, as stated in [22], apply to vector data in general, so they should not be limited to 3D point cloud techniques.
- It should be emphasized to which group the proposed algorithm belongs. Is it a form of storage feature-based watermarking because the watermarks are embedded by changing the relative storage order of the data, or zero watermarking because the evaluation compares the extracted watermark information with the copyright information registered with the copyright registration authority for similarity.
- Why is the evaluation of the "normalized correlation coefficient" (NC) performed with respect to the threshold set at 0.8, claiming that a watermark cannot be extracted at lower values? Please explain why the value is exactly 0.8?
- Table 2 conveys little information, as both CR and RMSE are 0 for both methods and both datasets. Perhaps suffice it to say that both algorithms compared are lossless and invisible.
- The text in Figure 9 needs to be much larger to be readable, and the readability of Figures 4 and 6 should also be improved.
- The sentence in lines 393 and 394 should be reworded, because the way it is written, one might think that the bit with the least weight is changed the least and the eighth bit from the end is changed the most, but it actually says that the bits from the last to the eighth bit from the end are changed.
- One should be careful with the claim that the watermark is not damaged by the application of projections, because the whole process is based on the Z coordinate of the point cloud, and the height is lost in the projection.
- The overall impression is that the manuscript is interesting, but that it is necessary to define more precisely and additionally explain some terms, values and reasons for their application. Also, it is very important to indicate up to what size of datasets and blocks it is reasonable to use the proposed technique and what is the performance.
- All the datasets used are extremely small. It should be stated why the proposed algorithm cannot or should not be applied to larger datasets, and the claims should be supported by numerical results, especially with respect to the time (and possibly memory) required to generate the watermark. The above also applies to vertical partitioning. How does the block size (group) affect performance?
Reviewer 3 Report
The authors design a lossless watermarking algorithm based on vertical stability for geographic point cloud data. The watermarking scheme cleverly utilizes a stable relative
storage order of vertical attribute and has been demonstrated well. Still, the authors are suggested to consider the follow points, which may enhance the quality of the work.
1. The authors should summarize the main contributions of this paper in Section 1.
2. The authors should use some more state-of-the-art techniques for lossless watermarking.
Few state of the art listed as-
https://doi.org/10.1145/3589761
https://doi.org/10.1016/j.sigpro.2020.107833
https://doi.org/10.1016/j.jisa.2020.102733
https://doi.org/10.1109/TEM.2021.3066090
https://doi.org/10.1016/j.compeleceng.2021.107255
3. The description in the article does not correspond to the figure. For example, Figure
2(a) should be Figure 3(a), in line 221 and 222.
4. Section 2.2 needs to be improved and explained comprehensively. In the section, there may be some writing errors, for example in line 210 and 211.
5. CR and RMSE are not very necessary for losslessness and invisibility because the authors’ algorithm only modifies the storage order.
Round 2
Reviewer 1 Report
All my concerns and questions are answered and applied into the manuscript after revision.
English quality is good. It is better to apply proofreading before publishing.